# New Vegetable Oils with Different Fatty Acids on Natural Rubber Composite Properties

**DOI:** 10.3390/polym13071108

**Published:** 2021-03-31

**Authors:** Siwarote Boonrasri, Pongdhorn Sae-Oui, Alissara Reungsang, Pornchai Rachtanapun

**Affiliations:** 1Faculty of Engineering and Agro-Industry, Maejo University, Chiang Mai 50290, Thailand; 2MTEC, National Science and Technology Development Agency (NSTDA), Pathumthani 12120, Thailand; pongdhor@mtec.or.th; 3Department of Biotechnology, Faculty of Technology, Khon Kaen University, Khon Kaen 40002, Thailand; alissara@kku.ac.th; 4Research Group for Development of Microbial Hydrogen Production Process, Khon Kaen University, Khon Kaen 40002, Thailand; 5Academy of Science, Royal Society of Thailand, Bangkok 10300, Thailand; 6School of Agro-Industry, Faculty of Agro-Industry, Chiang Mai University, Chiang Mai 50100, Thailand; 7Cluster of Agro Bio-Circular-Green Industry (Agro BCG), Chiang Mai University, Chiang Mai 50100, Thailand; 8Center of Excellence in Materials Science and Technology, Faculty of Science, Chiang Mai University, Chiang Mai 50200, Thailand

**Keywords:** vegetable oil, fatty acid, natural rubber composite

## Abstract

Owing to the toxicity of polycyclic aromatic (PCA) oils, much attention has been paid to the replacement of PCA oils by other nontoxic oils. This paper reports comparative study of the effects of new vegetable oils, i.e., Moringa oil (MO) and Niger oil (NO), on rheological, physical and dynamic properties of silica–filled natural rubber composite (NRC), in comparison with petroleum–based naphthenic oil (NTO). The results reveal that MO and NO exhibit higher thermal stability and better processability than NTO. Cure characteristics of the rubber compounds are not significantly affected by the oil type. It is also found that the NRCs containing MO or NO have better tensile strength and lower dynamic energy loss than the NRCs containing NTO. This may be because both MO and NO improve filler dispersion to a greater extent than NTO as supported by storage modulus and scanning electron microscopy results. Consequently, the present study suggests that MO and NO could be used as the alternative non–toxic oils for NRC without any loss of the properties evaluated.

## 1. Introduction

Natural rubber composite (NRC) was prepared by mixing natural rubber (NR) with reinforcing filler such as silica and/or carbon black to improve physical properties and dynamic properties of NR. However, the incorporation of high filler loading simultaneously requires the addition of plasticizers or oils such as petroleum–based oils to improve filler dispersion and reduce compound viscosity for easier processing. Plasticizers soften the compound by lubricating between rubber molecules and, thus, promote filler incorporation and dispersion during mixing. The presence of plasticizer therefore improves physical and dynamic properties of the NRC [1]. Among petroleum oils, distillate aromatic extract (DAE) oils have been commonly used as oils for rubber compounding in the tire industry [2]. However, it has been reported that DAE oils contain relatively high content of polycyclic aromatic hydrocarbons (PAHs). It is well known that many of these PAHs are identified or suspected as carcinogens having toxic effects on organisms of human [3]. Low-PAHs petroleum–based oils such as naphthenic oil (NTO) and treated distillate aromatic extract (TDAE) oil are recommended for DAE oil substitution. However, it is currently revealed that some petroleum–based clean oils may contain carcinogens [4]. Consequently, many attempts have been made to use various vegetable oils including their derivatives such as castor oil (CAO) [5], citrus oil (CIO) [6], coconut oil (CO) [7], linseed oil (LO) [8,9], rice bran oil (RBO) [10,11], rubber seed oil (RSO) [12,13], palm oil (PO) [14,15] and soybean oil (SO) [2] as plasticizers in carbon black–filled rubbers. Additionally, most research dealing with the utilization of green oils has been carried out in carbon black filled rubbers. Little attention is given to silica–filled NR compound, despite the great difference in surface activity of these fillers. Recently, it has been reported that Moringa oil (MO) and Niger oil (NO) contain relatively high oleic acid content (>70%) and linoleic acid content (>70%), respectively, which is interesting for many applications such as cosmetic, food, medicine, biopesticide and biodiesel [16,17,18]. However, there are no previous works reporting the effect of MO and NO on physical and dynamic properties of NRC. Since there is a trend in the application of green and nontoxic oils, it is of interest to carry out research investigating the potential use of new vegetable oils for rubber processing applications.

This research aims to study physical and dynamic properties of silica–filled NRC mixed with two vegetable oils (MO and NO) compared to petroleum–based oil (NTO). Morphology of the NRCs comprising various oils was also investigated and compared.

## 2. Materials and Methods

### 2.1. Materials

Natural rubber (STR 5L) used in this study was purchased from Thai Rubber Latex Corporation (Thailand) Public Co., Ltd. (Chonburi, Thailand). Bis–(3–triethoxysilylpropyl) tetrasulfide (TESPT) and precipitated silica (Tokusil–URT, average particle size of 30 nm and BET specific surface area of 170 m^2^/g) were supplied by JJ Degussa Co., Ltd. (Bangkok, Thailand) and Tokuyama Siam Silica Co., Ltd. (Rayong, Thailand), respectively. Zinc oxide and stearic acid were obtained from Chemmin Co., Ltd. (Samuthprakarn, Thailand). N–tert–butyl–2–benzothiazolesulfenamide (Santocure TBBS) was purchased from Reliance Technochem Co., Ltd. (Bangkok, Thailand) sulfur (S_8_) was supplied by Siam Chemical Public Co., Ltd. (Bangkok, Thailand). Lowinox© CPL, a butylated reaction product of p-cresol and dicyclopentadiene, was obtained from Lucky Four Co., Ltd. (Nonthaburi, Thailand). NTO, comprising 47% aliphatic content; 42% naphthenic content and 11% aromatic content, was obtained from Nynas Pte. Ltd., Singapore. MO and NO were supplied by Tea Oil and Plant Oils Development Center (Chiang Rai, Thailand). The fatty acid compositions of these vegetable oils are given in Table 1. Thermal stability of the oils used in this study was studied by thermogravimetric analyzer TGA, SDTA 851, Mettler Toledo, Columbus, Ohio, USA) using a heating rate of 20 °C/min under nitrogen atmosphere.

### 2.2. Compound Preparation and Testing

In this study, NRCs were prepared according to the formulations shown in Table 2 using a laboratory internal mixer (ChareonTUT: MX105–D40L50, ChareonTUT, Thailand) equipped with cam rotor. Mixing was carried out by using a two–step mixing method. In the first step, the mixing conditions were set as follows: rotor speed at 50 rpm, temperature at 90 °C and fill factor at 0.75. NR was initially masticated for 30 s prior to the addition of silane and half of silica. After mixing for 45 s, the residual silica and oil were added. After the total mixing time of 8 min, the compound masterbatch was obtained. In the second step, the rotor speed and temperature were set at 30 rpm and 50 °C, respectively. The masterbatch was mixed with TBBS, zinc oxide, stearic acid and sulfur for 3 min and then sheeted on a laboratory two–roll mill (LRM150 W, Labtech Engineering, Thailand) and kept at 25 °C for 1 day prior to testing.

After mixing, Mooney viscosity of the rubber compounds was measured by a Mooney viscometer (Visc–TECH++, Tech Pro, Columbia, Indiana, USA) at 100 °C in accordance with ISO 289-1. Cure characteristics of the rubber compounds were determined via a moving die rheometer (MDR, rheo TECHMD+, Tech Pro, Columbia, Indiana, USA) at 160 °C as per ISO 6502. Scorch time and cure time are represented by t_s_2 and t_c_95, respectively. Torque difference (Max. torque - Min. torque) was calculated and used to represent indirectly the crosslink density. Measurement of storage shear modulus (G’) as a function of strain of the rubber compounds was carried out using a rubber process analyzer (RPA 2000, Alpha Technology, Hudson, Ohio, USA) at frequency and temperature of 0.5 Hz and 100 °C, respectively. The rubber compounds were vulcanized by compression molding at 160 °C in an electrically–heated hydraulic press (G 30H Wabash Genesis Series Hydraulic Press, Wabash, Indiana, USA) according to their optimum cure time (t_c_95) obtained from the moving die rheometer (MDR). The vulcanized samples were stored at room temperature for at least 24 h before testing.

Degree of silica dispersion in NR matrix was investigated by using a field emission scanning electron microscope (FE–SEM, model S–4700, Hitachi, Tokyo, Japan) at 3 kV electron energy. The newly cryogenic fractured surfaces of the rubber specimens were coated with Pt–Pd prior to being examined.

Tensile properties were measured using a universal testing machine (model 5566; Instron corporation, Norwood, Massachusetts, USA) equipped with pneumatic clamp and extensometer, in accordance with ISO 37. The dumbbell (die Type 1) specimens were cut from the rubber vulcanized sheets having thickness of approximately 2 mm. The specimens were tested using a crosshead speed of 500 mm/min and a 1 kN load cell. The reported values of tensile properties were the average of 5 measurements. Damping factor (tan δ) was measured as a function of temperature by using a dynamic mechanical thermal analyzer; DMTA (Explexor TM 25 N, Gabo Qualimeter, Selb, Germany). Tan δ values at 60 °C and 0 °C which are directly related to rolling resistance and wet grip index of tires are reported. The test was performed in tension mode at a constant frequency of 10 Hz, static strain of 1%, and a dynamic strain of 0.1%. The temperature was scanned from –80 to 60 °C with a heating rate of 2 °C/min.

## 3. Results and Discussion

### 3.1. Thermal Stability of MO, NO and NTO

The thermal stability of vegetable oils and NTO was evaluated by TGA. The resulting TGA thermograms and decomposition temperatures are shown in Figure 1 and Table 3, respectively. It can be seen that NTO shows relatively poor thermal stability as proved by the lowest values of maximum decomposition rate temperature (T_max_) and onset temperature (T_onset_) among the oils studied. The decomposition temperature range is also broader than those of the vegetable oils. This might be due to the mixed composition of aromatic carbon, naphthenic carbon and paraffinic carbon in NTO. The vegetable oils, in contrast, display higher T_max_ and T_onset_ with narrower ranges of decomposition temperature than NTO. T_max_ values are found at 444 °C, 422 °C and 355 °C for MO, NO and NTO, respectively. NO has slightly lower T_max_ than MO due to the difference in fatty acid type.

### 3.2. Effect of Oils on Viscosity and Cure Characteristics

Figure 2 shows scorch time (t_s_2) and cure time (t_c_95) of the rubber compounds comprising different oil types and contents. The results show that both scorch time and cure time tend to increase with increasing oil content regardless of the oil type. This can be described by the dilution effect as oil could also consume the curatives. It is suggested that some portion of curatives might dissolve in oil phase and react with the unsaturated hydrocarbons in long hydrocarbon chains of the vegetable oils, which is analogous to the reduction in vulcanization efficiency. Moreover, as reported elsewhere, the oil can hinder the vulcanization reaction and reduce the crosslink density [25,26]. Therefore, the increases of scorch time and cure time with increasing oil content could be explained. Results also reveal that scorch time and cure time are independent of the oil type despite the differences in oil structures. 

Figure 3a shows the torque difference (Max. torque-Min. torque) which could be used to indicate indirectly the crosslink density of the rubber compounds comprising different oil types and contents. The crosslink density slightly decreases with increasing oil content regardless of the oil type. This could be explained by the curatives consumption of oils as previously mentioned. At a given oil content, the rubber compounds comprising vegetable oils exhibit slightly higher crosslink density than those comprising NTO. This cannot be readily explained at present. One of the possibilities is that NTO contains aromatic groups which can interfere with free radical reactions of vulcanization reaction while these aromatic groups are absent in the vegetable oils.

Mooney viscosities of the rubber compounds comprising different oil types and contents are displayed in Figure 3b. With increasing oil content, the compound viscosity continuously decreases. This might be due to the improved plasticization effect of oils used in this work as evidenced by the continuous reduction of glass transition temperature (Tg) with the increase of oil content as shown in Table 4.

### 3.3. Effect of Oils on Morphology and Storage Modulus

From the FE-SEM micrographs (Figure 4), it is found that all vegetable oils used in this work give similar degree of filler dispersion. Compared with NTO, both MO and NO give smaller size of silica agglomerates and, thus, provide better silica dispersion which leads to the significant improvements in the mechanical and dynamic properties [27,28,29,30].

The proposed model representing the interaction between the vegetable oils and silica is shown in Figure 5. The vegetable oils consist of polar ester groups which might interact with silanol groups on silica surfaces via hydrogen bond [31,32,33]. Thus, rubber-filler interaction is enhanced which would facilitate the dispersion of silica and reduce the filler–filler interaction [34]. Conversely, NTO comprises no polar group which can improve rubber-filler interaction. Hence, the poorer silica dispersion could be expected in the NTO-containing compounds [33].

Figure 6 displays the dynamic mechanical properties of the rubber compounds as determined by RPA. The results show the decrease of G’ with increasing strain for all oil types, representing the destruction of silica network with increasing strain [27,35]. Generally, the difference in G´ values at low and high strains (ΔG’) is used to indicate the degree of filler–filler interaction [33,36,37]. Apparently, due to the poorer silica dispersion, the NTO-containing compound has higher ΔG’ indicating a greater magnitude of filler–filler interaction.

### 3.4. Effect of Oils on Physical and Dynamic Properties

Elongation at break and 300% modulus of the NRCs are displayed in Figure 7. Obviously, modulus of the NRCs decreases with increasing oil content, while the opposite trend is found for elongation at break. This can be explained by the plasticization effect of oils. It is also observed that the modulus and elongation at break of all NRCs are slightly influenced by the oil type. However, the NRCs comprising NTO possess slightly lower modulus and elongation at break than the corresponding NRCs having vegetable oils, possibly due to the poorer filler dispersion [38,39].

Figure 8c presents the variations of tensile strength of the NRCs having different oil types and contents. Tensile strength increases with increasing oil content up to 6 phr and then slightly decreases afterwards. The initial improvement is thought to be the consequence from the improved silica dispersion (as illustrated in Figure 4). At higher oil contents, the plasticization effect becomes more dominant on the vulcanizate strength. At any given content, the NRCs containing vegetable oils give slightly greater tensile strength than those containing NTO, possibly due to the better filler dispersion as supported by the SEM results (see Figure 4) and ΔG′ results (see Figure 6). Similar observations have been reported in some published works in which the rubber compound having vegetable oil showed better tensile strength than that having petroleum–based oil [1,7,13]. Noticeably, both MO and NO give comparable tensile strength despite the differences in types and contents of fatty acid in their composition.

It is widely accepted that tan δ at 60 °C could be used to indicate the rolling resistance of tire tread, i.e., the lower the tan δ at 60 °C, the better the rolling resistance [34,40]. The results from Figure 8a reveal that the vegetable oils give better rolling resistance than NTO. The greater plasticization effect and the improved filler dispersion are used to explain such findings. Apart from the tan δ at 60 °C, tan δ at 0 °C is also used to indicate the wet grip index of tire tread, i.e., the higher the tan δ at 0 °C, the greater the wet grip index. The results in Figure 8b show that, at any given oil content, NTO gives higher value of tan δ at 0 °C than MO and NO indicating the superior wet grip index of NTO. This is not beyond expectation because both MO and NO have a greater plasticization effect and a greater ability to shift the glass transition temperature of the rubber towards lower temperatures. They therefore give lower values of tan δ at 0 °C. Taken as a whole, both Mo and No give a slight benefit on rolling resistance with a slight impairment of wet grip index when used in tire tread application, compared to NTO. The results reveal the potential application of these vegetable oils in the manufacturing of pneumatic tires when lower rolling resistance is required [41,42].

## 4. Conclusions

This study reveals that vegetable oils (MO and NO) provide many advantages over NTO in silica-filled NRCs. Cure characteristics of the NRCs are not significantly impacted by the oil type. Both vegetable oils give greater plasticization effect and better mechanical properties than NTO. This is because the vegetable oils provide better filler dispersion as supported by the ΔG’ and SEM results. In this study, tensile strength is found to reach a maximum at 6 phr of oil content. Dynamic properties are also improved as can be seen from the reduction of tan δ at 60 °C and, thus, rolling resistance when NTO is replaced by the vegetable oils. MO and NO give comparable NRCs properties despite the differences in types and contents of fatty acid in their composition.

## Figures and Tables

**Figure 1 polymers-13-01108-f001:**
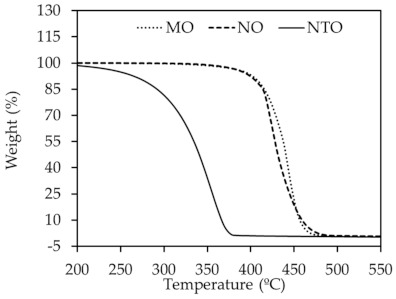
TGA curves of different oils.

**Figure 2 polymers-13-01108-f002:**
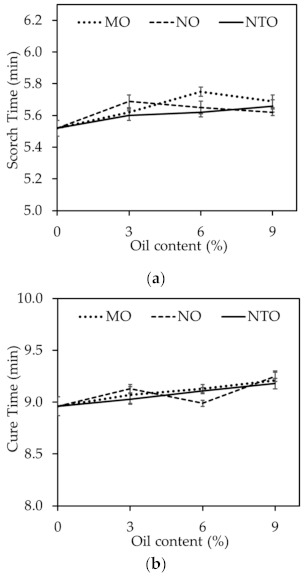
(**a**) Scorch time and (**b**) cure time of the rubber compounds comprising different oil types and contents.

**Figure 3 polymers-13-01108-f003:**
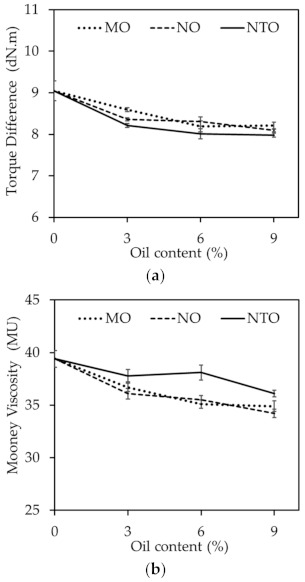
(**a**) Torque difference and (**b**) Mooney viscosities of the rubber compounds comprising different oil types and contents.

**Figure 4 polymers-13-01108-f004:**
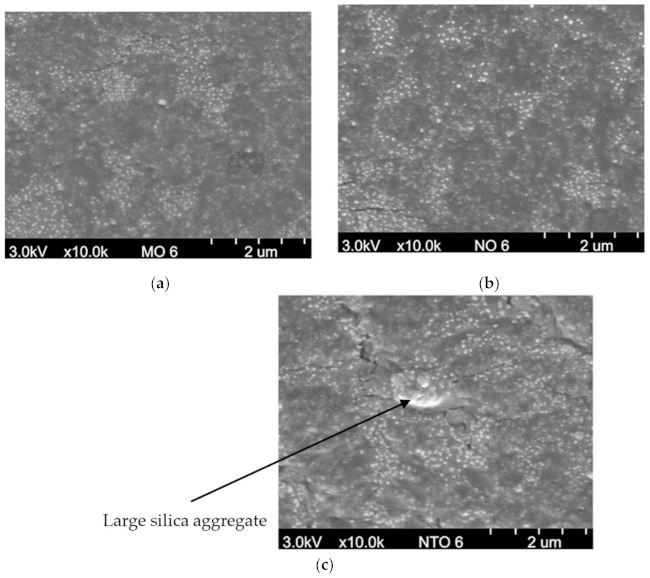
FE–SEM micrographs of the NRCs comprising 6 phr of oils (**a**) MO, (**b**) NO and (**c**) NTO.

**Figure 5 polymers-13-01108-f005:**
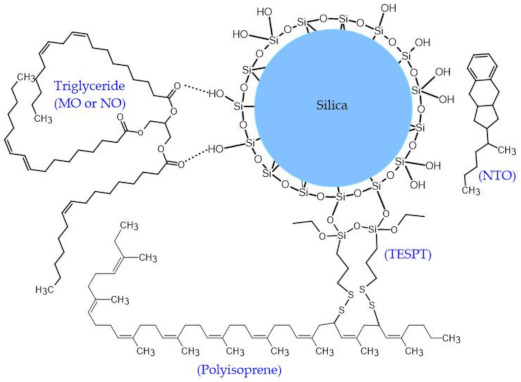
Proposed model illustration of the imaginable formation of hydrogen bonding between vegetable oils (MO and NO) and silica.

**Figure 6 polymers-13-01108-f006:**
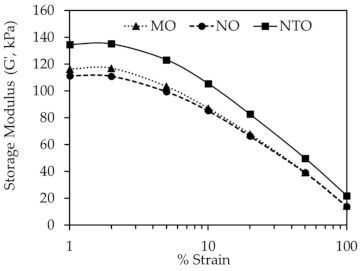
Storage modulus (G´) of the rubber compounds as a function of strain at 6 phr of different oils.

**Figure 7 polymers-13-01108-f007:**
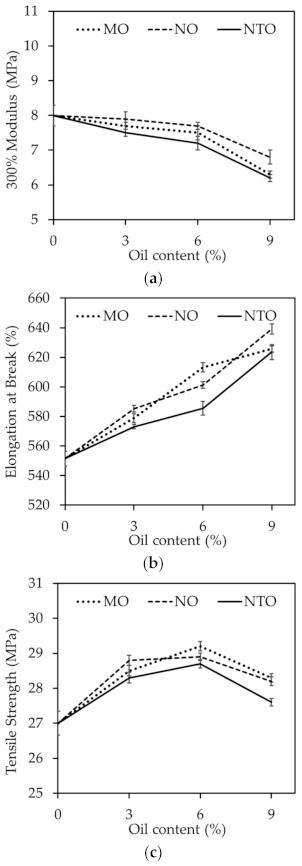
(**a**) 300% modulus and (**b**) elongation at break (**c**) tensile strength of the NRCs comprising different oil types and contents.

**Figure 8 polymers-13-01108-f008:**
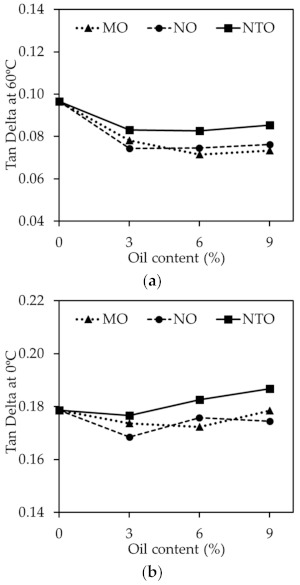
(**a**) tan δ at 0 °C and (**b**) tan δ at 60 °C of the NRCs comprising different oil types and contents.

**Table 1 polymers-13-01108-t001:** The fatty acid compositions of MO [19,20,21], NO [22,23] and NTO [24].

Composition	MO	NO	NTO
Palmitic acid (C16:0), %	7	8	-
Stearic acid (C18:0), %	6	6	-
Oleic acid (C18:1), %	74	16	-
Linoleic acid (C18:2), %	1	70	-
Other fatty acids, %	12	-	-
Aromatic content	-	-	11
Naphthenic content	-	-	42
Aliphatic content	-	-	47

**Table 2 polymers-13-01108-t002:** The compound formulations used in the present study.

Ingredients	Content, Part per Hundred of Rubber (Phr)
Natural rubber	100
ZnO	5
Stearic acid	3
Silica	43
TESPT	3
TBBS	2
Lowinox© CPL	1.5
Sulfur	2.2
Oil (MO, NO or NTO)	0, 3, 6, 9

**Table 3 polymers-13-01108-t003:** The decomposition temperatures of different oils studied.

Oil Type	Onset Temperature (T_onset_, °C)	Maximum Decomposition Rate Temperature (T_max_, °C)	Final Decomposition Temperature (T_final_, °C)	Decomposition Temperature Range (°C)	Residue at 700 °C (%)
MO	401	444	480	79	0.0
NO	402	422	485	83	0.0
NTO	251	355	371	120	0.0

**Table 4 polymers-13-01108-t004:** Glass transition temperature (Tg) of the rubber compounds plasticized with different oil types and contents.

Oil Type	Tg (°C)			
	0 phr	3 phr	6 phr	9 phr
MO	−43.8	−45.5	−46.0	−47.5
NO	−43.8	−45.4	−46.5	−47.3
NTO	−43.8	−45.2	−45.9	−46.3

## Data Availability

The data presented in this study are available on request from the corresponding author.

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
