# Peer review of "New Vegetable Oils with Different Fatty Acids on Natural Rubber Composite Properties"

_polymers, 2021, doi:10.3390/polym13071108_

Round 1

Reviewer 1 Report

The manuscript “New Vegetable Oils with Different Fatty Acids on Natural Rubber Composite Properties” by Boonsari and coworkers investigate the effect of using selected vegetable oils  compare with aromatic based oil on the properties of properties of silica–filled natural rubber composite (NRC).

Although the Manuscript provides a well-designed experimental set, it needs to be more oriented as a scientific paper than a technical report. Therefore:

  • The manuscript need to be revised and rewritten, in both sentences paraphrasing and scientific oriented discussion.
  • In thermal properties, TGA curve shows clear low, both T(onset and max), for NTO compared with MO and NO. However, there is no differences between MO and NO in their T(onset) but they have difference in their profile and T(max) by ~ 22 degree. This needs to be discussed in the manuscript. More discussion for the data in both Figure 1 and table 3 in needed.
  • Figures 3 needs more discussion. Similarity in Cure time for both MO and NTO needs to be highlighted and discussed.
  • Adding the heating/cooling curves for MO, NO and NTO to the manuscript with more discussion will be great.

Author Response

Response to Reviewer 1’s Comments

Point 1: The manuscript need to be revised and rewritten, in both sentences paraphrasing and scientific oriented discussion.

Response 1:  Correction has been made in the revised manuscript. (highlighted in red)

Point 2: In thermal properties, TGA curve shows clear low, both T(onset and max), for NTO compared with MO and NO. However, there is no differences between MO and NO in their T(onset) but they have difference in their profile and T(max) by ~ 22 degree. This needs to be discussed in the manuscript. More discussion for the data in both Figure 1 and table 3 in needed.

Response 2: NO has slightly lower Tmax than MO due to the difference in fatty acid type.   Discussion has been added on line 173, highlighted in red)

Point 3: Figures 3 needs more discussion. Similarity in Cure time for both MO and NTO needs to be highlighted and discussed.

Response 3:  Results also reveal that scorch time and cure time are independent of the oil type despite the differences in oil structures. Since vegetable oils are mainly complexes of triesters of glycerol called triglycerides. The long hydrocarbon chains in vegetable oils make them non-polar and not reactive to vulcanization reaction. The cure retardation found in this study therefore mainly arises from the dilution effect which is independent of the oil type. (Discussion has been added on lines 191-195, highlighted in red)

Point 4:  Adding the heating/cooling curves for MO, NO and NTO to the manuscript with more discussion will be great.  

Response 4: Unfortunately, we studied thermal stability of oils using TGA technique and, thus, it is not possible to have a cooling curve, unlike in DSC technique.

Reviewer 2 Report

The topic of the research is both interesting and oriented to the sustaibanle solutions and eco-friendly choise of plasticizers. The extent of testing is sufficient and the results are well-presented. Indeed, the findings can be beneficial for the fellow scientists working in the field. The manuscript is recommended for the publication after a minor revision.

  1. Please introduce the full plasticizer names in the introduction, not just abbreviations. The structure of the introduction must be changed - introduction to the topic first, as you did in the abstract. Justify, why these Mo and NO oils, which are mostly fatty acids, are compared to the naphtenic plasticizer oil except that it hasnt been studies before - they are quite different in structure, so why NTO was chosen. Why NR compound? Do you have any certain application in mind? So please extend and restructure the introduction.
  2. Please add the information about the silica, e.g. size. Could you explain more the selection of a recipe, any specific reason to use 43 phr of silica?
  3. MO and NO contain stearic and other fatty acids, which can affect the curing performance. What is your oppinion?
  4. Concerning the discussion of Fig 3a, can the interactions between selica and vegetable oils be related to the torque changes?
  5. Figure 5 is a nice illustration of possible interactions. Should you also mention it in relation to the silane?
  6. In the final parapraph of chapter 3 you mention tires as possible application for these plasticizers. In this case, loss tangent at 60oC isn't enough. Please provide the DMA data for 0oC, or add full DMA curves as a cupplementary data.

Author Response

Response to Reviewer 2’s Comments

Point 1: Please introduce the full plasticizer names in the introduction, not just abbreviations. The structure of the introduction must be changed - introduction to the topic first, as you did in the abstract. Justify, why these Mo and NO oils, which are mostly fatty acids, are compared to the naphtenic plasticizer oil except that it hasnt been studies before - they are quite different in structure, so why NTO was chosen. Why NR compound? Do you have any certain application in mind? So please extend and restructure the introduction.

Response 1: Correction has been made in the revised manuscript. (lines 57, 69 highlighted in red) Actually, we didn’t have any application in mind when we formulated the rubber compounds. We just wanted to compare the plasticization efficiency of these MO and NO with a low toxic and widely used mineral oil (that is why we selected naphthenic oil as a reference in this study) in the silica-filled compound. We selected NR because NR has long been used in tire and automotive industries for decades.

Point 2:  Please add the information about the silica, e.g. size. Could you explain more the selection of a recipe, any specific reason to use 43 phr of silica?

Response 2: Correction has been made in the revised manuscript. (lines 87 highlighted in red)  

Point 3: MO and NO contain stearic and other fatty acids, which can affect the curing performance. What is your opinion?

Response 3: Yes, we agree with the reviewer’s comment. Stearic acid and other fatty acids can affect the curing performance. However, in vegetable oils, the fatty acids react with glycerol and form “triglycerides” and, thus, their reactivity on cure behavior is not pronounced. In addition, most vegetable oils contain very low content of free fatty acids. Consequently, the addition of MO or NO does not give different cure characteristics, compared to the addition of NTO. 

Point 4:  Concerning the discussion of Fig 3a, can the interactions between silica and vegetable oils be related to the torque changes? 

Response 4: Yes. That could be another reason. Correction has been made in the revised manuscript accordingly. (line 205, highlighted in red) 

Point 5: Figure 5 is a nice illustration of possible interactions. Should you also mention it in relation to the silane?

Response 5: The authors agree with the reviewer’s comment. As the system contains both silane and vegetable oils which can react with the silanol groups on silica surface, the diagram in Figure 5 should be amended. We have included the silane-silica and vegetable oil-silica interactions into Figure 5 as kindly suggested. 

Point 6: In the final paragraph of chapter 3 you mention tires as possible application for these plasticizers. In this case, loss tangent at 60°C isn't enough. Please provide the DMA data for 0°C, or add full DMA curves as a supplementary data.

Response 6: Actually, these oils could be used to improve processability of filled rubbers in any applications. In this work, we just want to give the example of using these oils in tire tread application and, thus, we want to show the effect of these oils on rolling resistance of the rubber vulcanizates. As kindly suggested, we have included the results of tan delta at 0°C into the revised manuscript (lines 289-297, highlighted in red).

Reviewer 3 Report

The article is interesting, however, there are some features that should be improved.

Figures 2a and 2b could be put together in a single graph.

What type of clamp is used to be able to do the mechanical tests without the NR test samples draining before breaking

In materials and methods section is necessary to explain how the torque difference and Money viscosimetry are determined.

It is necessary to attach the characteristics of the SiO2 used.

The explanation about the relationship between the Tan d at 60% and rolling resistance should be improved.

I recommend strongly use FTIR technique in order to corroborate the possible interactions between the different oils used and the SiO2 particles as shown in the figure 5.

Conclusions are poor and needs to improve.

Author Response

Response to Reviewer 3’s Comments

Point 1: Figures 2a and 2b could be put together in a single graph.

Response 1:  We split the results into two graphs because we want to enlarge the Y-axis scale of each graph so that readers can see the differences between solid and dashed lines. If we put them into a single graph, it will be very difficult to see the differences between lines because the oil type has a very little effect on cure times.

Point 2: What type of clamp is used to be able to do the mechanical tests without the NR test samples draining before breaking

Response 2:  We used pneumatic clamp to grip the NR specimen and testing was carried out at relatively high crosshead speed. The clamp type has been added into the revised manuscript. (line 146, highlighted in red) 

Point 3: In materials and methods section is necessary to explain how the torque difference and Mooney viscosity are determined.

Response 3: Details of how to measure the torque difference and Mooney viscosity have been added into the revised manuscript. (lines 122 and 128, highlighted in red) 

Point 5: It is necessary to attach the characteristics of the SiO2 used.

Response 5: Basic properties of the silica used in this work have been added as shown on line 88 (highlighted in red).  

Point 6: The explanation about the relationship between the Tan d at 60% and rolling resistance should be improved.

Response 6:  More details of the relationship between tan delta at 60°C and rolling resistance have been added as shown on line 284-286

Point 7: I recommend strongly use FTIR technique in order to corroborate the possible interactions between the different oils used and the SiO2 particles as shown in the figure 5.

Response 7:  We agree with the reviewer’s comment that the proposed interaction between vegetable oils and silica, shown in Figure 5, needs some results or references to support the idea. Actually, we have tried to characterize the interaction by FTIR technique but, possibly due to the limitation of our FTIR, the results were not obvious. We therefore propose the possible interaction based on the information previously reported by other researchers as shown below. We have included these references into the revised manuscript.

The triglycerides in vegetable oils adsorb onto silica surfaces due to hydrogen bonding between the silanol groups and the ester carbonyl groups (Adhikari et al., 1994; Proctor et al., 1996 , Siwarote et al., 2017 ).

  1. Adhikari, C., Proctor, A., and Blyholder, G. D. Diffuse-reflectance fourier-transform infrared spectroscopy of vegetable oil triglyceride adsorption on silicic acid. J. Am. Oil Chem. Soc. 1994, 71, 589–594. doi: 10.1007/BF02540584
  2. Proctor, A., Adhikari, C., and Blyholder, G. D. Lipid adsorption on commercial silica hydrogels from hexane and changes in triglyceride complexes with time. J. Am. Oil Chem. Soc. 1996, 73, 693–698. doi: 10.1007/BF02517942
  3. Siwarote, B.; Sae-Oui, P.; Wirasate, S.; Suchiva, K. Effects of Bio-based Oils on Processing Properties and Cure Characteristics of Silica-filled Natural Rubber Compounds. J Rubber Res 2017, 20, 1-19, doi: 10.1007/bf03449138.

Point 8: Conclusions are poor and needs to improve.

Response 8:   Correction has been made in the revised manuscript. (lines 304-315, highlighted in red) 

Round 2

Reviewer 1 Report

The manuscript has been improved!

Author Response

Faculty of Agro-Industry,

Chiang Mai University

155 M.2, Mae Hia, Muang,

Chiang Mai, 50100

25th January 2021

Dear Editor,

We are pleased to resubmit for publication the minor revised version of Manuscript ID Polymers-1133404 entitled “New Vegetable Oils with Different Fatty Acids on Natural Rubber Composite Properties”. We appreciate the time and effort that academic editor has dedicated to providing your valuable feedback on our manuscript. We are grateful to the editor for his/her insightful comments on our paper.

We have carefully read proof the manuscript.

We believe we have attended to all the corrections suggested. Thank you for taking the time and energy to help us improve the paper.

Sincerely,

Assoc. Prof. Pornchai Rachtanapun

Reviewer 3 Report

Authors have modified the manuscript according to my recomendations. 

Author Response

(The authors gave the same response as above.)
